# Vagus Nerve Stimulation with Mild Stimulation Intensity Exerts Anti-Inflammatory and Neuroprotective Effects in Parkinson’s Disease Model Rats

**DOI:** 10.3390/biomedicines9070789

**Published:** 2021-07-07

**Authors:** Ittetsu Kin, Tatsuya Sasaki, Takao Yasuhara, Masahiro Kameda, Takashi Agari, Mihoko Okazaki, Kakeru Hosomoto, Yosuke Okazaki, Satoru Yabuno, Satoshi Kawauchi, Ken Kuwahara, Jun Morimoto, Kyohei Kin, Michiari Umakoshi, Yousuke Tomita, Naoki Tajiri, Cesario V. Borlongan, Isao Date

**Affiliations:** 1Department of Neurological Surgery, Okayama University Graduate School of Medicine, Dentistry and Pharmaceutical Sciences, Okayama 700-8558, Japan; ippachi1982@yahoo.co.jp (I.K.); tatu_tatu_sasa@yahoo.co.jp (T.S.); masahiro.kameda@nifty.com (M.K.); first202475@yahoo.co.jp (M.O.); k.hosomoto96@gmail.com (K.H.); sky_black_0413@yahoo.co.jp (Y.O.); satoru_yabuno0220@yahoo.co.jp (S.Y.); airngelion@gmail.com (S.K.); codomolive@gmail.com (K.K.); bxmsr585@gmail.com (J.M.); thekinkorea@gmail.com (K.K.); umakoshi1008@yahoo.co.jp (M.U.); tomitamiharayosu@gmail.com (Y.T.); idate333@md.okayama-u.ac.jp (I.D.); 2Department of Neurosurgery, Tokyo Metropolitan Neurological Hospital, Tokyo 183-0042, Japan; takashiagari@gmail.com; 3Department of Neurophysiology and Brain Science and Medical School, Graduate School of Medical Sciences and Medical School, Nagoya City University, Nagoya 464-0083, Japan; ntajiri@med.nagoya-cu.ac.jp; 4Department of Neurosurgery and Brain Repair, University of South Florida Morsani College of Medicine, 12901 Bruce B. Downs Blvd., Tampa, FL 33611, USA; cborlong@health.usf.edu

**Keywords:** anti-inflammation, less invasive therapy, new experimental device, Parkinson’s disease, vagus nerve stimulation

## Abstract

Background: The major surgical treatment for Parkinson’s disease (PD) is deep brain stimulation (DBS), but a less invasive treatment is desired. Vagus nerve stimulation (VNS) is a relatively safe treatment without cerebral invasiveness. In this study, we developed a wireless controllable electrical stimulator to examine the efficacy of VNS on PD model rats. Methods: Adult female Sprague-Dawley rats underwent placement of a cuff-type electrode and stimulator on the vagus nerve. Following which, 6-hydroxydopamine (6-OHDA) was administered into the left striatum to prepare a PD model. VNS was started immediately after 6-OHDA administration and continued for 14 days. We evaluated the therapeutic effects of VNS with behavioral and immunohistochemical outcome assays under different stimulation intensity (0.1, 0.25, 0.5 and 1 mA). Results: VNS with 0.25–0.5 mA intensity remarkably improved behavioral impairment, preserved dopamine neurons, reduced inflammatory glial cells, and increased noradrenergic neurons. On the other hand, VNS with 0.1 mA and 1 mA intensity did not display significant therapeutic efficacy. Conclusions: VNS with 0.25–0.5 mA intensity has anti-inflammatory and neuroprotective effects on PD model rats induced by 6-OHDA administration. In addition, we were able to confirm the practicality and effectiveness of the new experimental device.

## 1. Introduction

The main pathology of Parkinson’s disease (PD) is characterized by a loss of dopamine (DA) neurons in the substantia nigra pars compacta (SNc) associated with prolonged neuroinflammation [1]. Glial cell activation of microglia and astrocyte in the SNc stands as common features of both human PD patients and animal models of PD, playing vital roles in DA neuronal degeneration [1,2,3,4]. In postmortem analysis of PD patients, activated microglia, HLA-positive microglia [2], and increased density of astrocytes populated the SNc [3,5]. These proliferated and activated glial cells induce inflammatory mediators including tumor necrosis factor α (TNFα), interleukin 1β (IL-1β), IL-6, IFN-γ, and subsequently cause oxidative damage, altogether contributing to the acceleration of DA neuronal degeneration in the SNc [1,4,6,7].

PD onset and progression are caused not only by loss of DA neurons in the SNc but also by significant degeneration of noradrenalin (NA) neurons in the locus coeruleus (LC). Postmortem brains of PD patients showed remarkable degeneration of NA neurons in the LC as well as the loss of DA neurons in the SNc [8,9,10]. The LC serves as the largest source of NA production in the central nervous system with axonal projections to multiple sites in the whole brain. NA neurons in the LC also send direct projections to the striatum and SNc, protecting DA fibers and neurons and subsequently functionally regulating DA neurons in the SNc [11]. Consequently, the degeneration and loss of NA in the LC significantly contribute to DA neuronal degeneration in the SNc and to the progression of PD pathology [12,13]. In advanced PD patients treated with long-term administration of anti-PD drugs, deep brain stimulation (DBS) exhibits therapeutic effects. Despite the effectiveness of DBS, it has the crucial problem of cerebral invasiveness and this disadvantage might limit the application of DBS for PD. There is an underlying necessity for complementary and alternative surgical therapies to treat advanced PD patients with less invasiveness. Vagus nerve stimulation (VNS) is conducted without cerebral invasiveness. In the laboratory, VNS produces robust amelioration of chronic inflammatory and autoimmune disorders, primarily through its anti-inflammatory properties [14,15].

In this study, we assessed the neuroinflammation in PD by examining the anti-inflammatory effects of VNS. Recognizing the need for a clinically feasible approach in providing continuous electrical stimulation, we utilized a wireless controllable stimulation device. We evaluated the therapeutic effects of VNS with behavioral and immunohistochemical outcome assays, with emphasis on detailing the status of glial cells and NA neurons under different stimulation conditions.

## 2. Materials and Methods

### 2.1. Electrical Stimulation System for Animal Studies

We developed a wireless controllable electrical stimulation device named SAS-200 for animals with technical assistance from Unique Medical Co., Ltd. (Tokyo, Japan). SAS-200 was designed exceedingly miniaturized and light-weight. The size of the device is as follows: 40 × 20 × 20 mm (width × depth × height). The weight is 26 g including a lithium-ion battery (3.7 V, 250 mAh) in an aluminum chassis (Figure 1a–c). SAS-200 allows outputting biphasic pattern of square pulses, and to set 10 patterns of stimulation intensity ranging 0–2.0 mA, 11 patterns of duration ranging 0–300 Hz, 3 patterns of pulse width, and 5 patterns of stimulation cycle. Thus, SAS-200 consists of over 1500 patterns of stimulation conditions. SAS-200 receives a parameter command from control software for a Windows PC via Bluetooth dongle which is designed for this system. We can detect the battery residual amount and the signals for changing the stimulation parameter by the glimmering of the LED light located inside the device through the transparent screw (Figure 1d). The stimulation system consisted of SAS-200 as a stimulator, a standard Windows PC as a command post, and Bluetooth dongle for SAS-200 (Figure 1e). In the experiment, SAS-200 was fixed to the back of a rat with a suture through holes in the base. The battery was charged by a power supply from PC via USB cable and easily exchanged by loosening two screws at the lateral side of SAS-200 (Figure 1f). The greatest advantage of this system is that all rats received continuous electrical stimulation without anesthesia and moved freely during the experiment.

### 2.2. Electrode

We used cuff-type electrodes purchased from Unique Medical Co., Ltd. (Tokyo, Japan) (Figure 2a). Electrodes were composed of two curved silver wires (0.08 mm diameter) covered with a 4 mm section of polyethylene tubing (outer/inner diameter: 1.0 mm/0.5 mm). The silver wires were aligned 1.5 mm apart in parallel inside the cuff. A cut was made lengthwise along the tubing to allow the cuff to be wrapped around the nerve and then closed by a suture (Figure 2b). The insulation was removed to provide conductivity, allowing bipolar stimulation limited in surrounding the nerve. These electrodes were designed based on a similar method used by several previous studies [16,17,18].

### 2.3. Ethics Statement and Animals

All experimental procedures were conducted in accordance with Okayama University guidelines for animal experiments and were approved by the University’s committee on animal experimentation (examination protocol, #OKU-2017449 approved on 23 October 2017). Adult female Sprague-Dawley rats (SHIMIZU Laboratory Supplies Co., Ltd., Tokyo, Japan) weighing 200–250 g at the beginning of the study were used for all experiments. They were singly housed per cage in a temperature- and humidity-controlled room that was maintained on a 12 h light/dark cycle with free access to food and water. A total of 50 rats were used in this study. Ten rats were used for 5 groups, respectively. The data obtained from a total of 46 animals were used in the analysis, with the exclusion of 2 rats with more than 20% weight loss and of 2 dead rats.

### 2.4. Surgical Procedure

#### 2.4.1. VNS Surgery

All rats were anesthetized with a combination of three anesthetics intraperitoneally (i.p.) injected (0.3 mg/kg of medetomidine, 4 mg/kg of midazolam, and 5 mg/kg of butorphanol). Rats were then fixed in supine position on the heating pad at 37 °C. A 20 mm skin incision was made on the left ventral side of the neck. The sternohyoid and sternomastoid muscles were separated longitudinally and then retracted laterally until the carotid artery within the carotid sheath was exposed. The left vagus nerve was then carefully separated from the surrounding connective tissue. The left vagus nerve was surrounded by a cuff-type silver electrode, and both ends were closed with sutures. Thereafter, a small incision was made on the dorsal side of the neck, then a subcutaneous tunnel was made with the passer to allow the electrode terminal to be passed through. The electrode terminal was fixed at the dorsal neck with a suture, followed by connection to the SAS-200 external terminal. A rat and SAS-200 were covered with handmade jacket to keep the stimulator in a stable position during the experiment [19]. A control group of animals was subjected to the same surgery with a dummy pulse generator.

#### 2.4.2. 6-OHDA Lesioning

After the operation for VNS, rats were placed in a stereotaxic instrument (Narishige, Japan). Furthermore, 20 µg of 6-OHDA (4 µL of 5 mg/mL dissolved in saline containing 0.2 mg/mL ascorbic acid; Sigma-Aldrich Co. LLC, St. Louis, MO, USA) was injected into the left striatum with a 28 G Hamilton syringe. The lesion coordinates were as follows: 1 mm anterior to the bregma, 3 mm lateral to the sagittal suture, and 5 mm ventral to the surface of the brain with the tooth-bar set at −1.0 mm. The injection rate was 1 µL/min. After the injection, the syringe was left in place for additional 5 min, followed by being retracted slowly (1 mm/min).

### 2.5. VNS Stimulation Parameters and Experimental Protocol

Fifteen minutes after the 6-OHDA injection, VNS electrical stimulation commenced. We applied the conventional parameters for VNS used in clinical settings: cycle of 30 s on 5 min off, pulse frequency of 30 Hz, pulse width of 500 µs, for 14 consecutive days. Except for the stimulation intensity, these fixed parameters were identical for every rat in this study. Rats were randomly assigned into 5 groups, namely, control group and 0.1 mA, 0.25 mA, 0.5 mA, and 1 mA VNS groups. To evaluate the behavioral symptoms of PD, cylinder test and methamphetamine-induced rotation test were performed by blinded investigators on day 7 and day 14 after 6-OHDA lesioning. On day 7, the SAS-200 was removed from the body at the time of behavioral tests, and then re-fixed under anesthesia after the behavioral tests. Following behavioral tests on day14, animals were euthanized for immunohistochemical investigations. These experimental designs are shown in Figure 3.

### 2.6. Behavioral Tests

#### 2.6.1. Cylinder Test

We performed the cylinder test to assess the degree of forepaw asymmetry, on day 7 and day 14 after 6-OHDA injection. Rats were placed in a transparent cylinder (diameter: 20 cm, height: 30 cm) for 3 min. and the number of forepaw contacts to the cylinder wall was counted [20]. The score of the cylinder test was calculated as a contralateral bias: ((the number of contacts with the contralateral limb) − (the number of contacts with the ipsilateral limb)/(the number of total contacts) × 100)) [21,22,23]. In the cylinder test, asymmetry in forelimb use is evaluated to reveal spontaneous locomotor activity of unilateral dopamine-depleted rats [20].

#### 2.6.2. Methamphetamine-Induced Rotation Test

All rats were tested with intraperitoneal administration of methamphetamine (3 mg/kg, Dainippon Sumitomo Pharma, Osaka, Japan) on day 7 and day 14 after the cylinder test. Fifteen minutes after injection, the rotational behaviors were assessed for 90 min. with a video camera. Full 360° turns ipsilateral to the lesion were counted [22,23].

Methamphetamine-induced rotation test is used to evaluate the degree of preservation of functional DA neurons, imbalances in dopamine, and motor impairment caused by 6-OHDA lesion [24].

### 2.7. Fixation and Sectioning

On day 15, all rats were deeply anesthetized by injection of overdosed sodium pentobarbital (100 mg/kg). Rats were perfused with 150 mL of 4 °C phosphate-buffered saline (PBS) transcardially, followed by 150 mL of 4% paraformaldehyde (PFA) in PBS. Brains were carefully removed and incubated in 30% sucrose at 4 °C for 3 days.

For the preparation of cryostat sections, brains were frozen and stored at −70 °C. Six series of 40 µm-thick coronal brain sections including striatum and SNc were obtained with a cryostat, respectively. Brain sections were stored at −20 °C and used for immunohistochemical investigations [22,23].

### 2.8. Immunohistochemical Investigations

#### 2.8.1. Tyrosine Hydroxylase (TH) Immunohistochemical Investigations

Free-floating sections were blocked with 3% hydrogen peroxide in 70% methanol for 10 min. Sections were washed 3 times for 5 min. each time in PBS, followed by incubation overnight at 4 °C with rabbit anti-TH antibody (1:500; Chemicon, Temecula, CA, USA) with 3% normal horse serum. After several rinses in PBS, sections were incubated for 1 h in biotinylated donkey anti-rabbit IgG (1:500; Jackson Immuno Research Lab, West Grove, PA, USA), then for 30 min in avidin–biotin–peroxidase complex (Vector Laboratories, Burlingame, CA, USA). Subsequently, the sections were treated with 3,4-diaminobenzidine (DAB, Vector) and hydrogen peroxide, then mounted on albumin-coated slides and embedded with cover glass [22,23].

#### 2.8.2. Fluorescent Immunostaining of Microglias, Astrocyte and Noradrenaline Neurons

To explore the distribution of immunoreactive glial cells in neurons, astrocytes, and microglias, double immunofluorescence staining of glial fibrillary acidic protein (GFAP) and ionized calcium binding adaptor molecule 1 (Iba1) were performed. For analyzing the viability of NA neurons in the LC, double immunofluorescence staining of dopamine β hydroxylase (DβH) was performed. Sections of 40-μm-thickness of the striatum, SNc, and LC were used. The slices were washed 3 times with PBS, followed by incubation with 10% normal horse serum and primary antibodies; rabbit anti-GFAP antibody (1:1000; Novus Biologicals, Littleton, CO, USA) and rabbit anti-Iba1 antibody (1:250; Wako Pure Chemical Industries, Osaka, Japan) and rabbit-anti DβH antibody (1:500; Sigma-Aldrich Co. LLC, St. Louis, MO, USA) for 24 h at 4 °C, respectively. After rinsing in PBS, sections were incubated for 1 h in FITC-conjugated affinity-purified donkey anti-rabbit IgG (H + L) and 4,6-diamidino-2-phenylindole (DAPI; 2 drops/mL, R37606; Thermo Fisher, Waltham, MA, USA) in a dark chamber. The sections were then extensively washed with PBS and coverslipped. Both TH and fluorescent immunoreactivities were visualized using an inverted fluorescence phase-contrast microscope BZ-X710 (Keyence, Osaka, Japan).

### 2.9. Morphological Analyses

#### 2.9.1. TH Immunostaining

The optical density of TH-positive fibers in the striatum was determined and analyzed with a computerized analysis system as previously described [25]. Three sections at 0.5 ± 1.0 mm anterior to the bregma were randomly selected for quantitative analyses. The two areas adjacent to the needle tract of lesioned side and the symmetrical areas in the intact side were analyzed, respectively. The percentages of lesion to the intact side were evaluated in each section and the averages were used for statistical analyses. The images were computer-processed into binary images using an appropriate threshold (Image J; National Institutes of Health, Bethesda, MD, USA). Using Image J, we first defined the threshold of the TH-positive fibers on the lesion side, and then applied the same threshold to the intact side. Each area was then calculated for statistical analyses. The number of TH-positive neurons was counted of three sections at 4.8, 5.3 and 5.8 mm posterior to the bregma in the SNc, respectively. The number of TH-positive neurons was measured by manually counting the cell bodies. The percentage of the number of TH-positive neurons in the lesioned SNc to the intact side was analyzed and the average was used for the statistical analyses. For all 3 areas, both the lesion side and intact side of the striatum and SNc in each rat was calculated and analyzed [22,23].

#### 2.9.2. Activation of Microglia and Astrocytes in Striatum and SNc

The number of Iba1-positive cells and GFAP-positive cells in the lesion side of the striatum and SNc was counted in the two fixed areas (each 500 × 500 μm square) to evaluate glial reaction. Three different sections were randomly selected which were the same level corresponding to TH-immunostaining. In total, 6 representative areas for both the striatum and SNc were counted in each rat. Cell number averages were used for statistical analyses [23].

#### 2.9.3. Preservation of Noradrenergic Neurons in the LC

The density of NA neurons in the LC was analyzed. Three sections at 10 ± 1.0 mm posterior to the bregma were randomly selected for analyses. The percentage of the number of DβH-positive neurons in the lesioned LC to the intact side was analyzed and the average was used for the statistical analyses to evaluate preservation of the NA neurons. For all 3 areas, both the lesion side and intact side of the LC (each 700 × 500 μm) in each rat was analyzed and the average was used for the statistical analyses.

### 2.10. Statistical Analyses

All data were analyzed using SPSS ver. 20.0 software (SPSS, Chicago, IL, USA). To investigate statistical significance between multiple groups, one-way analysis of variance with subsequent Tukey’s tests were used. Statistical significance was preset at *p* < 0.05. The variance homogeneity was confirmed with Levene’s test. Mean values are presented with standard error (SE).

## 3. Results

### 3.1. Body Weight

All rats showed a decrease of body weight on day 7 and recovery on day 14 without significant differences (Figure 4). In the 0.5 mA and 1 mA VNS groups, one rat from each group lost more than 20% of its body weight during the experiment (sacrificed on day 7). In the 1 mA VNS group, two rats died on day 10 and day 12 with decreased body weight, respectively. In the control group, 0.1 mA, and 0.25 mA VNS groups, body weight was maintained within normal limits and no rat died during the experiment. In total, 4 rats were excluded from this study.

### 3.2. Behavioral Tests

#### 3.2.1. Cylinder Test

Rats in 0.25 mA and 0.5 mA VNS groups showed significant improvement on day 14 (contralateral bias: 21.4 ± 8.4 (0.25 mA) and 17.8 ± 8.6% (0.5 mA)), compared to control group (78.4 ± 5.3%: one-way ANOVA, F_4,41_ = 7.92 both *p* < 0.001) and 0.1 mA VNS group (65.2 ± 10.2%, both *p* = 0.013) (Figure 5A). Slight improvement was shown in 1 mA VNS group (44.4 ± 14.9%), but not in 0.1 mA VNS group (65.2 ± 10.2%) compared to control group.

#### 3.2.2. Methamphetamine-Induced Rotation Test

Rats in the 0.25 mA VNS group showed significant reduction in methamphetamine-induced rotations on day 14 (648 ± 149 turns/90 min) compared to control group (1541 ± 221 turns/90 min: one-way ANOVA, F_4,40_ = 3.91, *p* = 0.011) and 0.1 mA VNS group (1373 ± 182 turns/90 min, *p* = 0.049) (Figure 5B). The number of rotations on day 14 in the 0.5 mA (849 ± 179 turns/90 min) and 1 mA VNS groups (1089 ± 279 turns/90 min) slightly decreased, compared to the control group.

### 3.3. Immunohistochemical Investigations

#### 3.3.1. TH

Rats in 0.25 mA (29.5 ± 2.2%) and 0.5 mA (27.1 ± 2.9%) VNS groups showed significantly preserved density of TH-positive fibers in the striatum, compared to that in the control group (18.5 ± 2.2%; one-way ANO VA, F_4,41_ = 4.73, *p* = 0.0077 and *p* = 0.0462) (Figure 6a). Rats in the 0.1 mA (20.3 ± 1.5%) and 1 mA (21.2 ± 3.4%) VNS groups showed limited therapeutic effects on TH-positive fibers in the striatum.

TH-positive neurons in the SNc of rats in the 0.25 mA (58.2 ± 1.8%) and 0.5 mA (56.1 ± 2.7%) VNS groups was significantly preserved, compared to those in the control group (38.9 ± 3.0%; one-way ANOVA, F_4,40_ = 7.39, *p* < 0.001 and *p* = 0.0044) and in the 0.1 mA VNS group (42.3 ± 2.0%, *p* = 0.011 and *p* = 0.045) (Figure 6b). Rats in the0.1 mA and 1 mA (48.5 ± 3.6%) VNS groups showed non-significant effects on TH-positive neurons in the SNc.

#### 3.3.2. Iba1

In the 0.25 mA and 0.5 mA VNS groups, the number of Iba1-positive microglia significantly reduced both in the striatum (0.25 mA: 33.3 ± 1.8 cells/field, one-way ANOVA, F_4,20_ = 10.67, *p* < 0.001; 0.5 mA: 33.7 ± 1.1 cells/field, *p* < 0.001) and SNc (0.25 mA: 26.3 ± 0.9 cells/field, one-way ANOVA, F_4,20_ = 35.18, *p* < 0.001; 0.5 mA: 31.9 ± 1.8, *p* < 0.001), compared to the control group (striatum: 68 ± 7.4 cells/field; SNc: 50 ± 1.7) (Figure 7a,b). The suppressive effects against migrating microglia by 1 mA VNS group were significant in the SNc (43 ± 2.1 cells/field, *p* = 0.043) but not significant in the striatum (50 ± 5.8 cells/field, *p* = 0.08), compared to the control group. Moreover, the suppressive effects against migrating microglia by the 1 mA VNS group were significantly inferior to those in the 0.25 mA and 0.5 mA VNS groups (0.25 mA: *p* < 0.001, 0.5 mA: *p* = 0.004). Compared with the 0.1 mA VNS group in the SNc (48 ± 2.0 cells/field), the 0.25 mA and 0.5 mA VNS groups significantly suppressed migrating microglia (both: *p* < 0.001).

#### 3.3.3. GFAP

Rats in the 0.25 mA, 0.5 mA, and 1 mA VNS groups showed significant suppression of GFAP-positive cells both in the striatum (0.25 mA:17 ± 0.9 cells/field, one-way ANOVA, F_4,20_ = 48.62 *p* < 0.001; 0.5 mA:18 ± 0.6, *p* < 0.001, 1 mA:26 ± 1.1, *p* < 0.001) (Figure 8a) and SNc (0.25 mA:17 ± 1.2 cells/field, one-way ANOVA, F_4,20_ = 25.59, *p* < 0.001; 0.5 mA: 18 ± 0.9, *p* < 0.001, 1 mA:26 ± 1.2, *p* = 0.018), compared to the control (striatum: 33 ± 1.5 cells/field, SNc: 34 ± 2.4) (Figure 8b). Rats in the 0.25 mA and 0.5 mA VNS groups showed significant suppression of GFAP-positive cells both in the striatum and SNc, compared to the 1 mA VNS group (striatum: 0.25 mA and 0.5 mA, *p* < 0.001; SNc: 0.25 mA and 0.5 mA, *p* = 0.006 and *p* = 0.03). Compared to the 0.1 mA VNS group, the 0.25 mA and 0.5 mA VNS groups displayed significant suppression of GFAP-positive cells both in the striatum and SNc (all: *p* < 0.001).

#### 3.3.4. DβH

The density of NA neurons in LC was significantly increased in the 0.25 mA (29.5 ± 2.2%, one-way ANOVA, F_4,20_ = 19.6, *p* < 0.001), 0.5 mA (27.1 ± 2.9%, *p* < 0.001), and 1 mA (21.2 ± 3.4%, *p* = 0.038) VNS groups compared to the control group (18.5 ± 2.2%) and 0.1 mA VNS group (20.3 ± 1.5%) (Figure 9). The increased density of NA neurons in the 0.25 mA VNS group were significantly higher than those of the 1 mA VNS group (*p* = 0.038).

## 4. Discussion

In this study, we demonstrated the therapeutic effects of VNS (0.25 mA and 0.5 mA) on PD rats using a wireless controllable electrical stimulation device with several intensities to simulate clinical settings. The behavioral improvement and DA neuronal preservation in these animals may be due to anti-inflammatory effects and potentiation of increased DA/NA neuronal preservation induced by VNS.

### 4.1. VNS in Clinical Settings

VNS is an established treatment for refractory epilepsy, depression, and cluster headache [15,26,27]. Several basic and clinical studies demonstrated the therapeutic efficacy of VNS in ischemic stroke [28,29,30,31], cerebral hemorrhage [32], traumatic brain injury [16,33,34,35], migraine [36,37], and Alzheimer’s disease [38]. Beyond the central nervous system, VNS was also reported to provide therapeutic effects in systematic or local inflammatory disorder such as septic shock [39,40,41], acute myocardial infarction [42,43], acute lung injury [44], ileus [45], obesity [46,47], and rheumatoid arthritis [48]. For PD, DBS is usually performed safely, but intracranial intervention still has potential risks of intracranial hemorrhage and infection [49,50] with subsequent critical complications. VNS is relatively safe without intracranial manipulation and surgically performed with ease. Thus, the therapeutic potential of VNS for PD may become more apparent because of its enhanced safety and feasibility.

The vagus nerve plays multiple roles in homeostatic regulation of visceral functions. The vagus nerve is composed of 80% afferent sensory fibers that project upward from the viscera into the medulla and 20% efferent motor fiber that regulates visceral organs [14,15]. The vagus nerve is composed of three types of fibers: A fiber (large and myelinated), B fiber (mid-size and myelinated), and C fiber (small and unmyelinated) [51,52]. The right vagus nerve contains fibers to the sinus node that is a risk for bradycardia, so the left VNS generally tends to be favored as surgical targets. Although the exact mechanisms of anti-inflammatory effects of VNS are not completely understood, both afferent and efferent pathways via NA and acetylcholine may be involved [14,53]. Vagal afferents project to the nucleus of the solitary tract (NTS) which widely feeds to brainstem, forebrain structures, and both directly and indirectly linked to the LC leading to synergistic subsequent regulation of NA secretion [14,15]. VNS protects NA fibers in the LC via its afferent fibers with subsequent protection of DA neurons in the SN [54]. VNS modifies the electrical activity of LC while depletion of NA in the LC eliminated the effects of VNS [55]. In this study, we demonstrated that the therapeutic potential of VNS was accompanied by NA and DA neuronal preservation. In concert with a previous study, VNS increased locomotor activity in 6-OHDA-induced PD model rats, maintained striatal TH-positive fibers and nigral TH-positive cells, suppressed glial cell expression, and retained NA neurons in LC [54]. Our results paralleled these reported findings, but the stimulation system and conditions were different. We directly applied clinical VNS parameters used for epilepsy treatment and varied stimulation intensity to the PD model in this study. This is because, in addition to mimicking common parameters for actual clinical application, it has been reported to elevate NA in brainstem including LC [16,56]. Our results demonstrated that 0.25 mA and 0.5 mA conventional VNS for 14 consecutive days was effective. The antiepileptic effects of VNS are thought to be mediated by the elevation of NA in LC [57]. The direct application of this stimulation parameter possibly increased NA neurons in LC and DA neurons in SN, which may have direct therapeutic application in PD patients. In addition, long-term intermittent VNS will be more effective for PD patients, considering the neurodegenerative nature of the disease.

### 4.2. Anti-Inflammatory Effects of VNS

In addition, mild to moderate stimulation intensity of 0.25 mA and 0.5 mA remarkably suppressed the activation of microglia and astrocytes induced by 6-OHDA, but the very weak and strong stimulation intensity of 0.1 mA and 1 mA were less effective. Microglia and astrocyte have a major role to play in immune defense in the central nervous system. Activated glial cells release anti-inflammatory substances with neuroprotective effects. However, prolonged overactivation of glial cells produces inflammation mediator leading to DA neuronal degeneration in the SN [1,4,6,7]. Reduction in NA following LC degeneration leads to immune-overactivity of glial cells with consequent neurodegeneration [58,59,60]. In our study, the 0.25 mA and 0.5 mA VNS groups showed strong anti-inflammatory effects with suppression of the morphological change of glial cells. The anti-inflammatory effects of 0.25 mA and 0.5 mA VNS with subsequent therapeutic effects on DA/NA neuronal preservation were stronger than those of 0.1 mA and 1 mA VNS. Indeed, the VNS experiment examining the plasticity of motor cortex showed that moderate stimulation intensity was effective, while very weak and strong stimulation were less effective [61,62]. This phenomenon is called the inverted-u effect. It is hypothesized that the activation of A and B fibers, which have low current thresholds, is important for the anti-inflammatory effect of VNS [51,55,63]. C fibers are small fibers that compose most of the vagus nerve, but the activation of A and B fibers, which are lesser components of the vagus nerve, is especially important for the therapeutic effect of VNS [51,55,63]. We speculated that the inverted-u effect is due to the increased activity of A and B fibers caused by the mild-moderate intensity. As a possibility, a mild-moderate stimulation intensity of VNS (0.25 mA and 0.5 mA) demonstrated remarkable therapeutic efficacy via the inverted-u effect for LC, SN, and striatum as well. It was recently reported that VNS with high-intensity stimulation inhibited the onset of the treatment effect for a longer time and inhibited neuroplasticity [64]. Although the exact pathway may differ, our experiment also confirmed that high-intensity stimulation is unlikely to respond to treatment for PD. In addition, the non-intensity dependent therapeutic response may be due to the damage induced by higher intensity, imbalanced stimulation between afferent/efferent nerves, among other unknown mechanisms that require further investigation.

Although not statistically, the body weight decreased as the stimulation intensity increased. VNS reduced fat and contributed to body weight loss [46,47]. However, in this experiment, some rats died from the strong stimulation intensity of 1 mA despite the body weight loss of 20% or less. In this regard, the strong stimulation of VNS appears not to be suitable for PD treatment. The overall results might indicate that 0.25 mA and 0.5 mA VNS have increased NA neurons in LC, nurturing NA to protect DA neurons in SN by suppressing glial cell activation with consequent attenuation of behavioral PD symptoms and of DA neuronal degeneration.

### 4.3. PD Pathogenesis and Inflammation

The pathological process of PD may originate from inflammation of peripheral nerve system in visceral organs, and progresses through the vagus nerve to the brainstem and SNc like a prion mechanism [65]. This inflammation-based hypothesis has become one of the mainstreams of the cause and progression of PD. The association between abnormalities in the intestinal flora and PD onset implicates key intestinal infections as the cause of PD. In tandem, abnormal intestinal flora may lead to amplified inflammatory flora and decreased anti-inflammatory flora, causing chronic constipation and ultimately a risk of developing PD [66]. Abnormal α-synuclein produced by glial cells in the intestinal tract propagate through the vagus nerve, reach the NTS to LC, and propagate to the SN leading to the development of PD [67]. Indeed, a recent large cohort study found that PD was suppressed in patients who had previously undergone abdominal vagotomy for gastric ulcer [52] or appendectomy [68]. The abnormal signals from the intestine via the vagus nerve may be suppressed. Although the exact mechanisms are still unclear, we recognized that the inflammatory pathogenesis of PD acts as a key target to treat PD. To this end, VNS may sequester this inflammation-plagued neurodegeneration in PD via its anti-inflammatory effects. If PD onset involves the spread of inflammation and neurodegeneration from the periphery to LC and SN through the vagus nerve pathway, cervical VNS may directly alter this pathway in providing relief for PD. In our study, VNS was started immediately after 6-OHDA administration. An early VNS initiation may suppress the glial cell activation and maintain the potency of LC-NA and SN-DA signaling, thereby preventing the progression of PD symptoms in its initial stages.

### 4.4. Study Limitations

We have several limitations in this experiment. We used a two-week-stimulation with a single condition without stepwise adjustment which is usually employed in the clinic. During the long-term treatment, the gradual increase of the stimulation intensity may enhance more neuroprotective effects. In addition, considering the pathophysiology of PD, it would be more ideal to perform VNS at 2–4 weeks after 6-OHDA administration. Further studies of late intervention against the PD model are needed to simulate clinical settings. The therapeutic mechanisms of VNS may require the participation of either afferent, efferent, in combination, stimulation, which will be explored in subsequent experiments including research for cholinergic anti-inflammatory pathways via efferent nerve and dynamics of inflammatory cytokines in the tissue. In addition, recent reports have suggested that the right vagus nerve has a greater number of TH positive nerves [69], and the right VNS is important in influencing DA regulation in the SN [70]. It is necessary to try the right VNS in PD research as well, instead of being limited to the traditional left VNS. In addition, although we performed PD treatment by varying the stimulation intensity in four steps, we need to conduct further searches by changing the stimulation intensity more closely, and further experiments by changing the intervals and frequency of stimulation. For example, as 1% of the stimulation intensity used for treating epilepsy in humans has neural plasticity on cortical lesion in rats [18,61,62]. Moreover, low frequency stimulation should be tried, as it predominantly activates the efferent fibers [14]. Additionally, most recently, the efficacy of burst VNS with high-frequency stimulation was reported to be effective for PD treatment in rats [71]. Further research is needed to determine what stimulation patterns are most effective in which areas of the brain. Since SAS-200 can create more than 1500 different stimulation parameters, additional experiments using these parameters may support us to discover appropriate stimulation parameters for PD treatment.

## 5. Conclusions

VNS with mild to moderate intensity exerted anti-inflammatory and neuroprotective effects on the PD rat model induced by 6-OHDA administration. These effects accompanied the preservation of DA neurons in nigrostriatal systems, increased NA neurons in LC, and robust behavioral improvements. The clinical entry of VNS emphasizes the crucial role of regulating the inflammation in the pathogenesis of PD. However, further studies of late intervention against the PD model are needed to simulate clinical settings. In addition, the usability and effectiveness of the new experimental device is hopeful for subsequent researches with electrical stimulation.

## Figures and Tables

**Figure 1 biomedicines-09-00789-f001:**
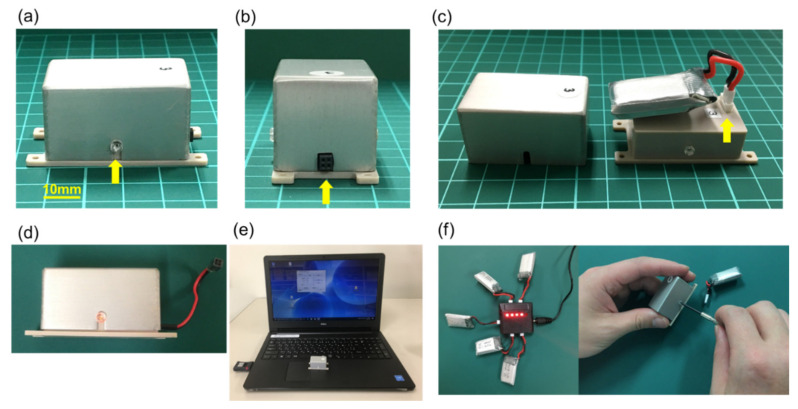
Newly developed stimulating system. (**a**) The lateral side of the SAS-200. A yellow arrow shows the transparent screw in which the LED light is attached to the electronic circuit board to confirm the energization. (**b**) The frontal side of the SAS-200. A yellow arrow is showing the external terminal of the SAS-200 where the stimulation electrode is connected. The external terminal can be extended with a lead type connection. (**c**) The lithium-ion battery in the aluminum chassis. The lithium-ion battery is connected to the control board inside the plastic chassis (arrow). (**d**) Flickering of the LED light located inside of SAS-200 through the transparent screw can indicate the stimulation command input and battery residual. The extensional lead can be attached to SAS-200. (**e**) SAS-200, Windows PC, and Bluetooth dongle for SAS-200. The recent Windows PC adapted with SAS-200 and its dongle. (**f**) Charging lithium-ion battery via USB cable. The battery can be exchanged by loosening two screws on the lateral side of SAS-200.

**Figure 2 biomedicines-09-00789-f002:**
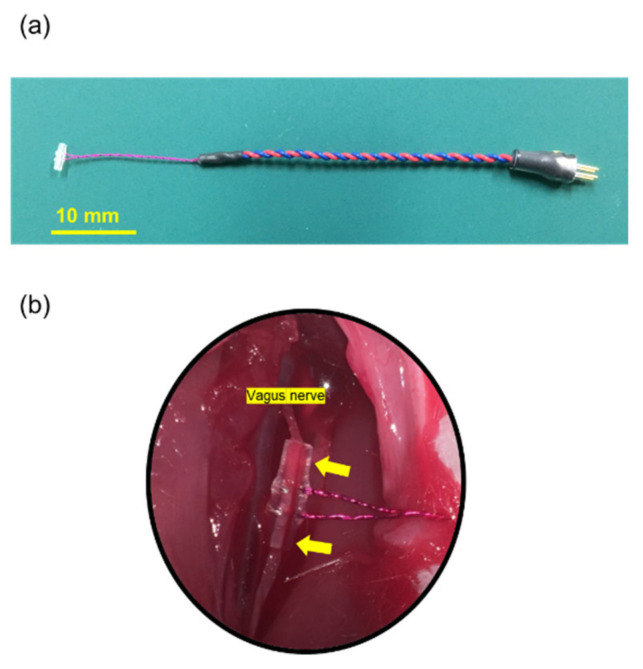
The cuff type electrode for vagus nerve stimulation. (**a**) The cuff type electrode used in this study. The electrode is 60 mm long, composed of 20 mm of silver wire and 40 mm of flexible lead. (**b**) The cuff type electrode surrounding the left vagus nerve. A yellow arrow shows the point to be fixed by 5–0 silk.

**Figure 3 biomedicines-09-00789-f003:**
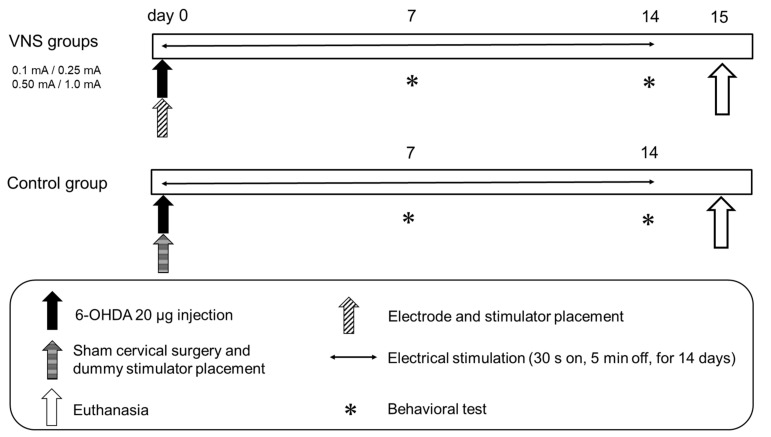
Experimental protocol. Fifty rats were randomly divided into five groups (*n* = 10 in each). On day 0, the left vagus nerve was exposed, and the cuff type silver electrode was placed. The electrode lead was connected to the SAS-200 and fixed to the back of the rats. Thereafter, 20 μg of 6-hydroxydopamine (6-OHDA) was stereotactically administered into the left striatum to prepare a Parkinson’s disease (PD) model. Vagus nerve stimulation (VNS) was started immediately after 6-OHDA administration and was continued for 14 days. On days 7 and 14, behavioral evaluation was performed with cylinder test and methamphetamine-induced rotation test. All rats were then euthanized on day 15.

**Figure 4 biomedicines-09-00789-f004:**
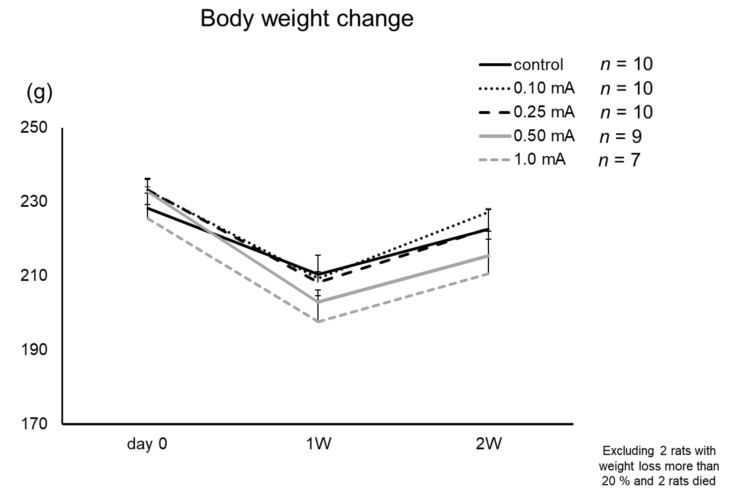
Body weight changes. Body weight decreased slightly with moderate intensity (0.1–0.25 mA) stimulation. In the 0.5 mA VNS group, one rat lost over 20% weight during the experiment (euthanized at day 7). In the 1 mA VNS group, one rat lost over 20% weight (euthanized at day 7) and two rats died before 2 weeks. Four rats were excluded from this study.

**Figure 5 biomedicines-09-00789-f005:**
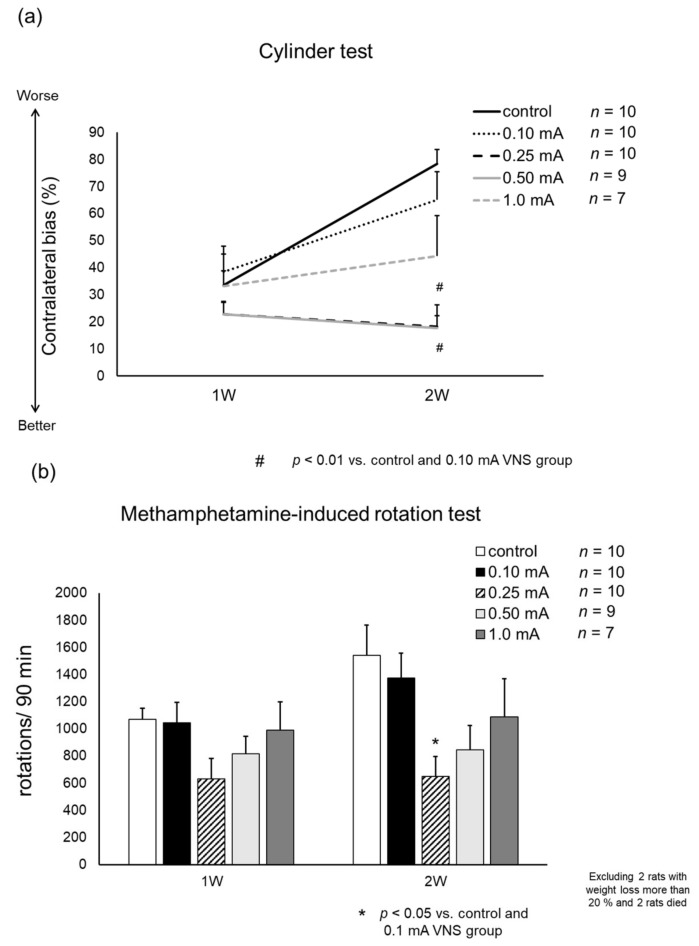
Results of behavioral tests. (**a**) The results of contralateral bias in the cylinder test. In the 0.25 mA and 0.5 mA VNS groups, the improvement of contralateral bias was observed at days 7 and 14 (^#^
*p* < 0.01 at day 14 vs. control and 0.1 mA VNS groups). (**b**) The number of methamphetamine-induced rotations per 90 min. The number of methamphetamine-induced rotations decreased in 0.25 mA and 0.5 mA VNS groups compared to the control group on days 7 and 14. The 0.25 mA VNS group significantly decreased rotation number at day 14 (* *p* < 0.05 vs. control and 0.1 mA VNS group). The data are presented as means ± SE and analyzed by one-way ANOVA and Turkey’s post hoc tests. *n* = 7–10 rats/group.

**Figure 6 biomedicines-09-00789-f006:**
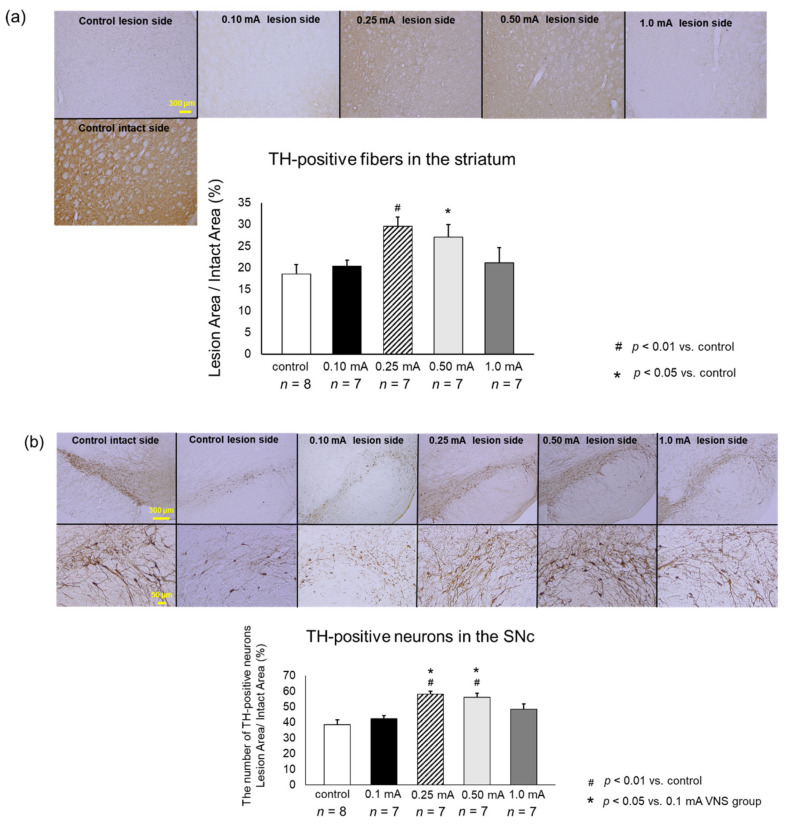
VNS with mild to moderate stimulation intensity preserved tyrosine hydroxylase (TH)-positive fibers in the striatum and TH-positive neurons in the SNc. (**a**) TH-positive fibers in the striatum. The ratio of TH-positive fibers in the lesioned striatum to the intact side was significantly preserved in the 0.25 mA and 0.5 mA VNS groups compared to that in the control group (^#^
*p* < 0.01 vs. control, * *p* < 0.05). (**b**) TH-positive neurons in the SNc.TH-positive neurons in the SNc in 0.25 mA and 0.5 mA VNS groups were significantly preserved compared to those in the control group and 0.1 mA VNS group (^#^
*p* < 0.01 vs. control, * *p* < 0.05 vs. control). The data are presented as means ± SE and analyzed by one-way ANOVA and Turkey’s post hoc tests. *n* = 7–8 rats/group.

**Figure 7 biomedicines-09-00789-f007:**
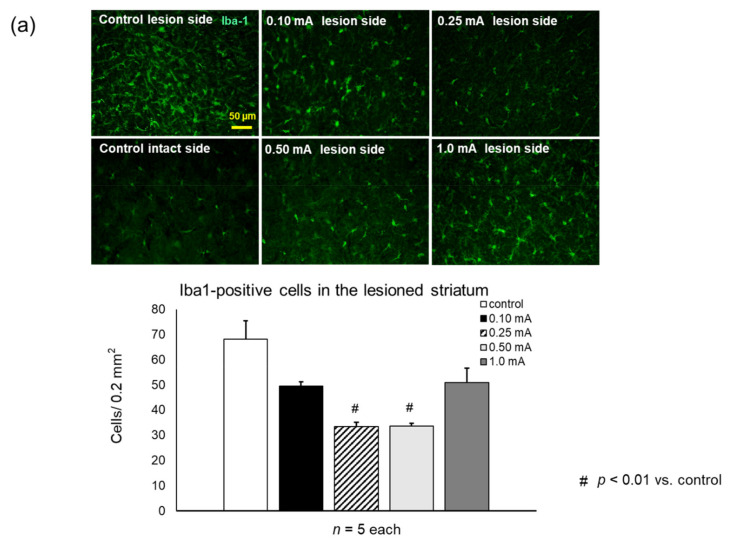
VNS with mild to moderate stimulation intensity inhibited proliferation of microglia both in the striatum and SNc. (**a**) Ionized calcium binding adaptor molecule 1 (Iba1)-positive microglia in the striatum of the lesion side. The number of Iba1-positive microglia in the lesioned SNc significantly decreased in the 0.25 mA and 0.5 mA VNS groups compared to the control group (^#^
*p* < 0.01). (**b**) Iba1-positive microglia in the SNc of the lesion side. The number of Iba1-positive microglia in the lesioned SNc significantly decreased in the 0.25 mA and 0.5 mA VNS groups compared to the control and 0.1 mA VNS groups (^#^
*p* < 0.01). Iba1-positive neurons in the SNc in the 1 mA VNS group also significantly decreased compared to the control group (* *p* < 0.05), although the effect was less than that of the 0.25 mA and 0.5 mA VNS groups. The data are presented as means ± SE and analyzed by one-way ANOVA and Turkey’s post hoc tests. *n* = 5 rats/group.

**Figure 8 biomedicines-09-00789-f008:**
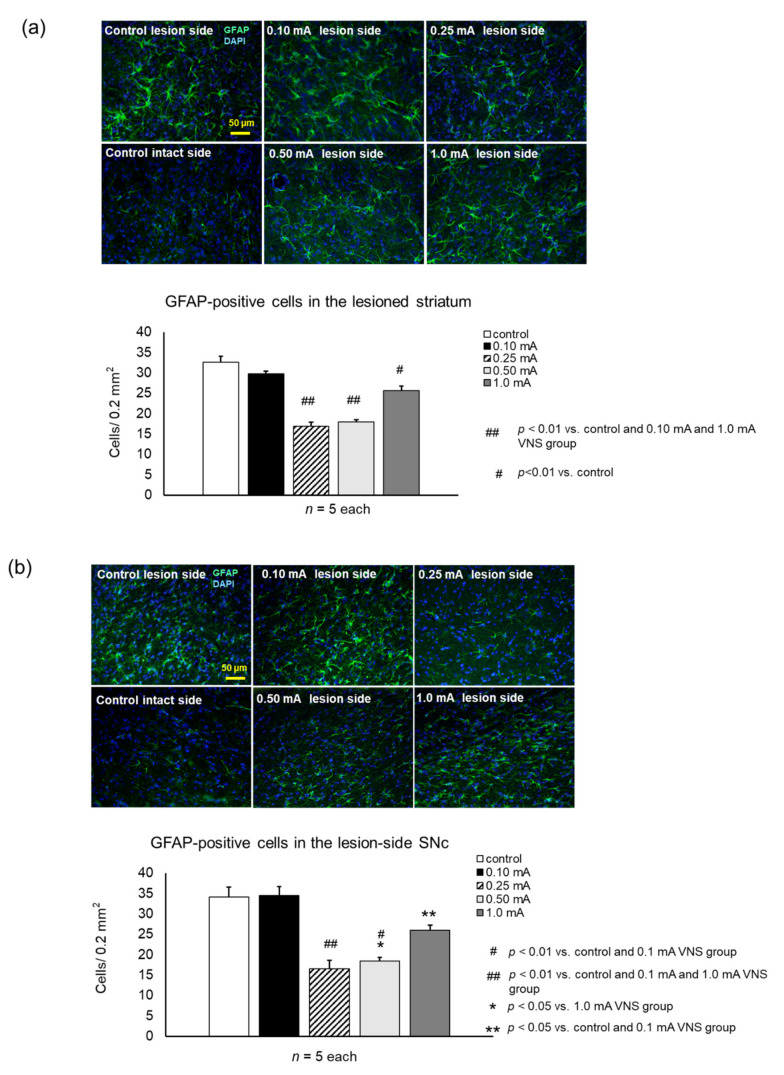
VNS with mild to moderate stimulation intensity inhibited astrocytic proliferation both in the striatum and SNc. (**a**) Glial fibrillary acidic protein (GFAP)-positive astrocytes in the striatum of the lesion side. The number of GFAP-positive astrocytes in the lesioned striatum significantly decreased in the 0.25 mA and 0.5 mA VNS groups to all other groups (^##^
*p* < 0.01). GFAP-positive astrocytes also significantly decreased in the 1 mA VNS group, compared to the control group (^#^
*p* < 0.01), although the effect was less than that of the 0.25 mA and 0.5 mA VNS groups. (**b**) GFAP-positive astrocytes in the SNc of the lesion side. The number of GFAP-positive astrocytes in the lesioned SNc significantly decreased in the 0.25 mA, 0.5 mA, and 1 mA VNS groups. Especially, 0.25 mA and 0.5 mA VNS groups displayed remarkable effects. Furthermore, the 0.25 mA and 0.5 mA VNS groups reduced astrocytes compared to all the other groups (^##^
*p* < 0.01 in 0.25 mA VNS group, ^#^
*p* < 0.01 and * *p* < 0.05 in the 0.5 mA VNS group). The 1 mA VNS group also reduced astrocytes compared to the control and 0.1 mA VNS groups (** *p* < 0.05). The data are presented as means ± SE and analyzed by one-way ANOVA and Turkey’s post hoc tests. *n* = 5 rats/group.

**Figure 9 biomedicines-09-00789-f009:**
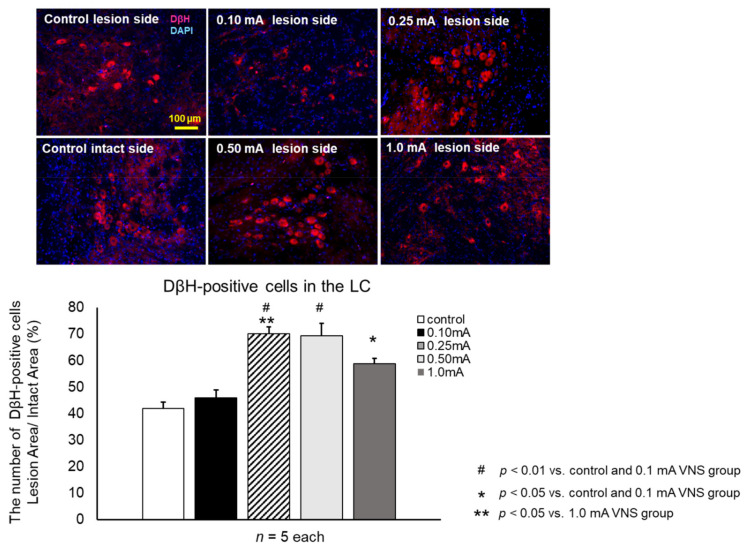
VNS with mild to moderate stimulation intensity preserved noradrenergic neuron in the LC. The ratio of dopamine β hydroxylase (DβH)-positive noradrenergic neurons in the lesioned LC to the intact side was significantly increased in the 0.25 mA, 0.5 mA, and 1 mA VNS groups compared to the control and 0.1 mA VNS groups (^##^
*p* < 0.01 in 0.25 mA VNS group, ^#^
*p* < 0.01 in 0.5 mA VNS group, ** *p* < 0.05 in 1 mA VNS group). In addition, the preservation effect of the 0.25 mA VNS group was significantly superior to that of the 1 mA VNS group (* *p* < 0.05). The data are presented as means ± SE and analyzed by one-way ANOVA and Turkey’s post hoc tests. *n* = 5 rats/group.

## Data Availability

The data presented in this study are available on request from the corresponding author.

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
