# Peer review of "Vagus Nerve Stimulation with Mild Stimulation Intensity Exerts Anti-Inflammatory and Neuroprotective Effects in Parkinson’s Disease Model Rats"

_biomedicines, 2021, doi:10.3390/biomedicines9070789_

Round 1

Reviewer 1 Report

Reviewer's evaluation

Manuscript ID: biomedicines-1278404

Title: Vagus nerve stimulation with mild stimulation intensity ameliorates behavioral and immunohistochemical investigations in Parkinson’s disease model rats
Journal: Biomedicines

Authors: Ittetsu Kin, Tatsuya Sasaki, Takao Yasuhara *, Masahiro Kameda,
Takashi Agari, Mihoko Okazaki, Kakeru Hosomoto, Yosuke Okazaki, Satoru
Yabuno, Satoshi Kawauchi, Ken Kuwahara, Jun Morimoto, Kyohei Kin, Michiari
Umakoshi, Yousuke Tomita, Naoki Tajiri, Cesario V. Borlongan, Isao Date

The authors of the peer-reviewed paper developed an animal model for treating Parkinson's disease (PD) with vagus nerve stimulation as a possible alternative to more traumatic deep brain stimulation. It has been shown that stimulation of the vagus nerve with strictly defined parameters in 6-OHDA-treated rats (PD model) is associated with a decrease in neuroinflammation, as well as with neuroprotection of dopaminergic neurons of the nigrostriatal system and noradrenergic neurons of the locus coeruleus, which are involved in the regulation of motor behavior and degenerate in PD. This exposure finally led to an improvement in motor behavior. In addition to obtaining the above results, the undoubted advantage of this study is the development of a miniature vagus nerve stimulator. Along with the rather positive impression of the paper, it raises a number of questions and comments.

  1. The title of the paper needs to be changed. Indeed, "Vagus Nerve Stimulation with Mild Stimulation Intensity..." cannot “ameliorate immunohistochemical investigations”, since immunohistochemistry is a tool, not a result. Apparently, the authors mean such promising results as neuroprotection and a decrease in gliosis.

The same correction needs to be made in a number of phrases, such as: "The improved behavioral and immunohistochemical PD symptoms in these animals may be due to anti-inflammatory effects and potentiation of increased DA / NA neuronal preservation induced by VNS." Indeed, there is no such notion as "immunohistochemical PD symptoms" in neurology.

  1. The fact that the vagus stimulation begins immediately after the single administration of the neurotoxin means that the authors have developed a model of preventive therapy at the preclinical stage of PD, i.e. before the onset of motor symptoms, which are used to diagnose PD. What are the reasons to believe that this therapy will also be effective at the clinical stage of PD, i.e. many years after the onset of the disease with degeneration of at least 50-60% of nigrostriatal dopaminergic neurons? What models of PD / approaches could be used to solve this issue?
  2. It should be clarified why, with a single injection of 6-OHDA, the stimulation of the vagus was carried out for 14 days, despite the fact that the period of neurodegeneration in response to acute administration of neurotoxin is usually limited to 3-5 days.
  3. The authors state “In this study, we firstly demonstrated the therapeutic effects of VNS (0.25mA and 382 0.5mA) on PD rats using the wireless controllable electrical stimulation device with several intensities to simulate clinical settings.” It is desirable to clarify what kind of progress has been made in this work in comparison with previous similar studies, primarily with that of Farrand et al. (2020).
  4. It is necessary to clarify how the number of TH-immunoreactive axons in the striatum was assessed - by counting the number of individual axons and axonal varicosities or by estimating the optical density of TH-immunostaining in the selected zones or otherwise.
  5. The final conclusion should be formulated more clearly, indicating that stimulation of the vagus nerve has anti-inflammatory and neuroprotective effects in rats on the 6-OHDA PD model.
  6. It is desirable to clarify whether this work makes an essential contribution to the understanding of the mechanisms of anti-inflammatory and neuroprotective effects of vagus stimulation showed by the authors. More specifically, whether these effects are the result of direct action of the vagus nerve on the brain or are they mediated through the action of the vagus in the periphery, for example, on the gastrointestinal tract. It is also desirable to learn the authors’ opinion, which approaches could be used in the future for differential assessment of the impact of vagus stimulation on the brain - direct or indirect through the periphery.
  7. In some places, authors' claims should be supported by references to original research. For example: "... NA neuronal degeneration in the LC precedes the loss of DA neurons in the SNc in PD progression." Considering that 50-60% of nigrostriatal dopaminergic neurons are already degenerated when PD is diagnosed by motor symptoms, it is difficult to imagine obtaining evidence of the above statement using pathological material or PET-scan before diagnosing of PD by the appearance of motor symptoms.
  8. In the Material and Methods section, it is not indicated that between incubation of the brain in 30% sucrose and preparation of cryostat sections, the brain was frozen, apparently at 40-42°C and stored at -70°C.

Thus, this paper can be reconsidered for publication after major revision.

Author Response

Response to Reviewers’ comments

We wish to express our appreciation to the editors and reviewers for their insightful comments, which have helped us to significantly improve our manuscript. Our point-by-point responses to the Reviewers’ comments are detailed below. We submit two documents, highlighted version of all changes and incorporated clean version of revised manuscript. The page and line number refer to the incorporated version.

Reviewer 1

The authors of the peer-reviewed paper developed an animal model for treating Parkinson's disease (PD) with vagus nerve stimulation as a possible alternative to more traumatic deep brain stimulation. It has been shown that stimulation of the vagus nerve with strictly defined parameters in 6-OHDA-treated rats (PD model) is associated with a decrease in neuroinflammation, as well as with neuroprotection of dopaminergic neurons of the nigrostriatal system and noradrenergic neurons of the locus coeruleus, which are involved in the regulation of motor behavior and degenerate in PD. This exposure finally led to an improvement in motor behavior. In addition to obtaining the above results, the undoubted advantage of this study is the development of a miniature vagus nerve stimulator. Along with the rather positive impression of the paper, it raises a number of questions and comments.

Response: Thank you for your supportive comments. The point-by-point responses are described below.

  1. The title of the paper needs to be changed. Indeed, "Vagus Nerve Stimulation with Mild Stimulation Intensity..." cannot “ameliorate immunohistochemical investigations”, since immunohistochemistry is a tool, not a result. Apparently, the authors mean such promising results as neuroprotection and a decrease in gliosis.

The same correction needs to be made in a number of phrases, such as: "The improved behavioral and immunohistochemical PD symptoms in these animals may be due to anti-inflammatory effects and potentiation of increased DA / NA neuronal preservation induced by VNS." Indeed, there is no such notion as "immunohistochemical PD symptoms" in neurology.

Response: Thank you for your kind suggestion. We changed the title of this manuscript to  "Vagus Nerve Stimulation with Mild Stimulation Intensity Exerts Anti-inflammatory and Neuroprotective Effects in Parkinson's Disease Model Rats“.

We also corrected the phrases like "immunohistochemical PD symptoms" to the appropriate expression in the whole text. (Title, Abstract, Discussion)

  1. The fact that the vagus stimulation begins immediately after the single administration of the neurotoxin means that the authors have developed a model of preventive therapy at the preclinical stage of PD, i.e. before the onset of motor symptoms, which are used to diagnose PD. What are the reasons to believe that this therapy will also be effective at the clinical stage of PD, i.e. many years after the onset of the disease with degeneration of at least 50-60% of nigrostriatal dopaminergic neurons? What models of PD / approaches could be used to solve this issue?

Response: Thank you for your great and difficult question. The question you pointed out always remain with this experimental protocol. In this study, we made PD model and immediately after that, therapeutic intervention started. This is an experimental protocol based on previous PD studies in our laboratory (Shinko A.2014, Sasaki T 2016, Kuwahara K.2020).

We think that the therapeutic potentials of the intervention are more likely to be shown strongly in this protocol. The novel therapeutic options, which we do not know whether this treatment is effective for PD or not, might be good to be started with this type of protocol. After confirmation of the therapeutic potentials, we need to move to the next experiment to show whether the treatment is enough effective for advanced PD model.

However, as you kindly pointed out, the protocol is just a mimic of treatment for PD patients at a very early stage, but not for advanced PD patients. It does not reflect the clinical settings. We need to modify the expression in our manuscript and to add description on this issue as study limitation. Now the description was a little toned down. We should not definitively conclude that it has a therapeutic effect on advanced PD at this time point. Additional experiments with late treatment are needed to clarify the therapeutic effects on advanced PD. Considering the pathophysiology of PD induced by 6-OHDA, it would be more ideal to perform VNS at 2-4 weeks after 6-OHDA administration.

Considering the clinical features of PD, we believe that therapeutic intervention for transgenic rodent models or rodent models administered with alpha-synuclein protein is also preferable. There are several examples of α-synuclein models, including a model using fibrils extracts of Lewy bodies from post-mortem human tissue, and a model using viral vectors with α-synuclein-encoding genes (Konnova EA. 2018). We would like to reveal the efficacy of VNS using these models in the future. (p.16, -P.17 l.10)

  1. It should be clarified why, with a single injection of 6-OHDA, the stimulation of the vagus was carried out for 14 days, despite the fact that the period of neurodegeneration in response to acute administration of neurotoxin is usually limited to 3-5 days.

Response: Thank you for your comments. If 6-OHDA is injected in substantia nigra or medial forebrain bundle, the toxin directly affects the neural cells (complete lesion). However, as Dr. Lee (Dr. Bjőrklund’s team) reported in 1996, if 6-OHDA is injected into the striatum, same as in our study, toxin is taken in the striatal terminal and transferred to nigral cell body with gradual degeneration of nigral cells (partial lesion). In the experiment using partial lesion by 6-OHDA, the degeneration continues at least 2-4 weeks. Like many laboratories, we have performed partial lesion PD model for over 20 years. Recently, we performed 2week-electrical stimulation (spinal cord stimulation) experiments for PD models with partial lesion (Shinko A et al .2014, Kuwahara K et al.2020). In previous study of VNS on PD model at other lab, stimulation has been continued for 10 days (Farrand et al .2017). We consider that 2week stimulation experiment is appropriate for PD study using partial lesion.

  1. The authors state “In this study, we firstly demonstrated the therapeutic effects of VNS (0.25mA and 0.5mA) on PD rats using the wireless controllable electrical stimulation device with several intensities to simulate clinical settings.” It is desirable to clarify what kind of progress has been made in this work in comparison with previous similar studies, primarily with that of Farrand et al. (2020).

Response: Thank you for your thoughtful comments. In this study, we developed a novel electrical stimulator and its control system to perform continuous electrical stimulation. Formerly, wired electrical stimulation experiments on animals were mainstream, but there were several problems in anesthesia, motion limitation, skin problems, and so on. This new stimulator is detachable, which gives the animals fewer restrictions on behavior, and it was effective for long-term electrical stimulation. This system enables us to minimize the psychological effects and stress level of rats. It is also useful to use a Windows PC to change parameters via Bluetooth.

The device used in the latest study by Farrand et al. (2020) is similar to our concept. However, it is different from our device because the stimulation conditions were fixed and couldn’t be adjusted. We can control the stimulation conditions (strength, duration, and pattern) as required in various situations. With our system, various stimulation settings are usable for other experiments. In their study, stimulation frequency and pattern were changed with fixed stimulation intensity. In our study, we were the first to change the stimulation intensity in VNS experiments on PD.

However, as you kindly pointed out, the word “firstly” might mislead the readers of this paper. We deleted the word “firstly” from this manuscript.

Again, we emphasize that this stimulation system is useful for experiments of electrical stimulation on animals because it can create over 1500 conditions that are close to actual clinical practice. Currently, we are using this stimulation system to conduct multiple electrical stimulation experiments (SCS, VNS, and DBS) simultaneously. We will report the results of these experiments in another report in the future. (p.14 l.5 from the bottom)

  1. It is necessary to clarify how the number of TH-immunoreactive axons in the striatum was assessed - by counting the number of individual axons and axonal varicosities or by estimating the optical density of TH-immunostaining in the selected zones or otherwise.

Response: Thank you for your constructive suggestion. We used the same method of previous studies to evaluate the density of TH-positive fibers in the striatum (Kadota et al. 2009, Shinko A et al. 2014, Kuwahara K et al.2020). We added the detailed description to the original Method section as follows.

The optical density of TH-immunoreactive fibers in the striatum was assessed by using Image J. The two areas adjacent to the needle tract of lesioned side and the symmetrical areas in the intact side were analyzed, respectively. Using image J, we first define the threshold of the cell fibers on the lesion side, and then apply the same threshold to the intact side. The percentages of lesion to the intact side were evaluated in each section and the averages were used for statistical analyses. Every 3 area for both the lesion side and intact side of the striatum in each rat was calculated and analyzed. (p.7 2.9.1)

  1. The final conclusion should be formulated more clearly, indicating that stimulation of the vagus nerve has anti-inflammatory and neuroprotective effects in rats on the 6-OHDA PD model.

Response: Thank you for your kind suggestion. We revised the final conclusion to make it clear in accordance with your suggestion. (Abstract, Conclusion)

  1. It is desirable to clarify whether this work makes an essential contribution to the understanding of the mechanisms of anti-inflammatory and neuroprotective effects of vagus stimulation showed by the authors. More specifically, whether these effects are the result of direct action of the vagus nerve on the brain or are they mediated through the action of the vagus in the periphery, for example, on the gastrointestinal tract. It is also desirable to learn the authors’ opinion, which approaches could be used in the future for differential assessment of the impact of vagus stimulation on the brain - direct or indirect through the periphery.

Response: Thank you for your important comments. To be honest, we have only partly understood the anti-inflammatory effects of vagus nerve stimulation in this experiment.

We have not yet reached a clear conclusion on whether the effect of VNS is owing to afferent or efferent stimulation. Based on recent studies, we believe that VNS has both effects(Johnson RL et al. 2018), although the dominancy of afferent/efferent is still unknown.

Because the vagus nerve is the largest cranial nerve and widely distributed in the body. To understand completely the anti-inflammatory effect of VNS requires extensive experiments. Our experiment is like a groundbreaking result, which is the first step toward the clinical application of VNS for PD.

At now, in our laboratory, we are performing unilateral vagotomy with stimulating either afferent or efferent fibers independently to investigate their effects on the brain/viscera. As the reviewer kindly suggested, the next experiment is needed. The data is now being collected.

Furthermore, other experiments on VNS gave us suggestions on this issue. Reversing the cathode and anode in bipolar stimulation (Ahmed U 2020) and electrically blocking nerve fibers with high frequency stimulation over 1000 Hz (Patel YA 2016) might be another key experiment for us.

(Oriiginally described in 4.4 Study limitation)

  1. In some places, authors' claims should be supported by references to original research. For example: "... NA neuronal degeneration in the LC precedes the loss of DA neurons in the SNc in PD progression." Considering that 50-60% of nigrostriatal dopaminergic neurons are already degenerated when PD is diagnosed by motor symptoms, it is difficult to imagine obtaining evidence of the above statement using pathological material or PET-scan before diagnosing of PD by the appearance of motor symptoms.

Response: Thank you for your suggestion. We explained this statement with a direct quote from the following article (Paredes-Rodriguez E et al. 2020) They described that degeneration of the LC precedes degeneration of the DA.

However, as you kindly point out, it is still difficult to clinically measure degeneration of the LC. This expression is just a hypothesis. In order to avoid misleading, the expression was now omitted from the manuscript. (Introduction, p.2 from the first)

  1. In the Material and Methods section, it is not indicated that between incubation of the brain in 30% sucrose and preparation of cryostat sections, the brain was frozen, apparently at 40-42°C and stored at -70°C.

Response: Thank you for your suggestion. We added this explanation to the Methods section. (Methods, p.6 2.7)

Reviewer 2

The authors describe an elegant study that tests the effects of vagus nerve stimulation on behavioural outcomes and neuroinflammation in a rat model of Parkinson’s Disease. Different, clinically relevant stimulation intensities were tested, and it was found that moderate stimulation, rather than the lowest or highest intensities tested proved most beneficial, resulting in decreased inflammatory glial cells and increased noradrenergic neurons. I believe that the study is of interest and contributes to the field and should be accepted for publication following minor revisions and review to improve the language and grammar.

The manuscript could be improved by addressing the following points:

Response: Thank you for your supportive comments. The point-by-point responses are described below.

  1. In the Materials and Methods (Page 4, Line 126), the authors state that the experiments resulted in ‘2 dead rats’. Greater explanation is required to explain this, did they die during the surgical procedures (eg. As a consequence of anaesthetic overdose), or following surgical recovery while receiving VNS. Was a post-mortem performed? If so, what were the results.

Response: Thank you for your suggestion. Two rats died maybe due to the strong stimulation of 1mA on Day10 and 12 with decreased body weight, respectively. We added this explanation to the manuscript. We did not perform a postmortem evaluation on the two rats. We considered that the rats died of consumption by 1mA stimulation, but we should have performed postmortem evaluation in order to investigate the cause of death. (p.8 3.1)

  1. In the described study, the authors used 2 behavioural tests, the cylinder test and methamphetamine-induced rotation test. Given that there a range of other tests that can be applied (eg. T-maze, treadmill running test), it would be beneficial to add a sentence or 2 describing what these particular tests are evaluating in the Parkinson’s disease model eg. Spontaneous locomotor activity, imbalances in dopamine etc.

Response: Thank you for your kind suggestion. We added the following contents in our manuscript.

  1. In the cylinder test, asymmetry in forelimb use is evaluated to reveal spontaneous locomotor activity of unilateral dopamine-depleted rats. (p.8 3.2.1)
  2. Methamphetamine-induced rotation test is used to evaluate the degree of preservation of functional DA neurons, imbalances in dopamine, and motor impairment caused by 6-OHDA lesion. (p.8 3.2.2)

  1. The authors present non-significant statistical analyses in their results (eg. Page 8, line 304-305). Generally, only significant results are reported, with the subsequent data and p values presented, while it is simply stated that other results were not significant. The authors may wish to remove the additional information from the Results.

Response: Thank you for your comment. We removed unnecessary data from the Results.

(Results, many places such as p.9, 3.1)

  1. Labels and scale bars on some of the figures are difficult to read. For example, the scale bars in Figure 6, Iba-1 in Figure 7 etc. The authors should make these labels more legible.   

Response: Thank you for your kind comment. We edited the scale bar and improved the figure quality. (Figures)

Again, thank you for giving us the opportunity to strengthen our manuscript with your valuable comments and queries. We have worked hard to incorporate your feedback and hope that these revisions persuade you to accept our submission.

Sincerely yours,

Takao Yasuhara

Department of Neurological Surgery, Okayama University Graduate School of Medicine, Dentistry and Pharmaceutical Sciences, 2-5-1, Shikata-cho, Kita-ku, Okayama-shi, Okayama 700-8558, Japan

Tel.: +81-86-235-7336; fax: +81-86-227-0191

E-mail address: [email protected]

Reviewer 2 Report

The authors describe an elegant study that tests the effects of vagus nerve stimulation on behavioural outcomes and neuroinflammation in a rat model of Parkinson’s Disease. Different, clinically relevant stimulation intensities were tested, and it was found that moderate stimulation, rather than the lowest or highest intensities tested proved most beneficial, resulting in decreased inflammatory glial cells and increased noradrenergic neurons. I believe that the study is of interest and contributes to the field and should be accepted for publication following minor revisions and review to improve the language and grammar.

The manuscript could be improved by addressing the following points:

  • In the Materials and Methods (Page 4, Line 126), the authors state that the experiments resulted in ‘2 dead rats’. Greater explanation is required to explain this, did they die during the surgical procedures (eg. As a consequence of anaesthetic overdose), or following surgical recovery while receiving VNS. Was a post-mortem performed? If so, what were the results.
  • In the described study, the authors used 2 behavioural tests, the cylinder test and methamphetamine-induced rotation test. Given that there a range of other tests that can be applied (eg. T-maze, treadmill running test), it would be beneficial to add a sentence or 2 describing what these particular tests are evaluating in the Parkinson’s disease model eg. Spontaneous locomotor activity, imbalances in dopamine etc.
  • The authors present non-significant statistical analyses in their results (eg. Page 8, line 304-305). Generally, only significant results are reported, with the subsequent data and p values presented, while it is simply stated that other results were not significant. The authors may wish to remove the additional information from the Results.
  • Labels and scale bars on some of the figures are difficult to read. For example, the scale bars in Figure 6, Iba-1 in Figure 7 etc. The authors should make these labels more legible.   

Author Response

(The authors gave the same response as above.)

Round 2

Reviewer 1 Report

The authors thoroughly followed the reiewer's recommendations. Some concerns were clarified, whereas others are considered targets for future reserach.

Thus, the paper can be accepted for publication.